# Risk of Recurrent Peptic Ulcer Disease in Patients Receiving Cumulative Defined Daily Dose of Nonsteroidal Anti-Inflammatory Drugs

**DOI:** 10.3390/jcm8101722

**Published:** 2019-10-18

**Authors:** Chih-Ming Liang, Shih-Cheng Yang, Cheng-Kun Wu, Yu-Chi Li, Wen-Shuo Yeh, Wei-Chen Tai, Chen-Hsiang Lee, Yao-Hsu Yang, Tzu-Hsien Tsai, Chien-Ning Hsu, Seng-Kee Chuah

**Affiliations:** 1Division of Hepato-Gastroenterology, Department of Internal Medicine, Kaohsiung Chang Gung Memorial Hospital and Chang Gung University College of Medicine, Kaohsiung 83300, Taiwan; gimy54861439@gmail.com (C.-M.L.); d5637700@cgmh.org.tw (S.-C.Y.); eggclimb@cgmh.org.tw (C.-K.W.); tintinjohn1220@gmail.com (Y.-C.L.); superthankyou1006@gmail.com (W.-S.Y.); luketai1019@gmail.com (W.-C.T.); 2Division of Infectious Diseases, Department of Internal Medicine, Kaohsiung Chang Gung Memorial Hospital and Chang Gung University College of Medicine, Kaohsiung 83300, Taiwan; lee900@cgmh.org.tw; 3Department of Traditional Chinese Medicine, Chiayi Chang Gung Memorial Hospital, Chiayi 83300, Taiwan; r95841012@ntu.edu.tw; 4Health Information and Epidemiology Laboratory of Chang Gung Memorial Hospital, Chiayi 61363, Taiwan; 5School of Traditional Chinese Medicine, College of Medicine, Chang Gung University, Taoyuan 33302, Taiwan; 6Division of Cardiology, Department of Internal Medicine, Kaohsiung Chang Gung Memorial Hospital and Chang Gung University College of Medicine, Kaohsiung 83300, Taiwan; garytsai@cgmh.org.tw; 7Department of Pharmacy, Kaohsiung Chang Gung Memorial Hospital, Kaohsiung 83300, Taiwan; 8School of Pharmacy, Kaohsiung Medical University, Kaohsiung 80700, Taiwan

**Keywords:** nationwide study, nonsteroidal anti-inflammatory drugs, *Helicobacter pylori* eradication therapy, recurrent peptic ulcer disease

## Abstract

The purpose of this population-based case–control study was to clarify the impact of cumulative dosage of nonsteroidal anti-inflammatory drugs (NSAIDs) on recurrent peptic ulcers among chronic users after *Helicobacter pylori (H. pylori)* eradication. We analyzed data of 203,407 adult peptic ulcer disease (PUD) patients from the National Health Insurance Research Database in Taiwan entered between 1997 and 2013. After matching for age/gender frequencies and the length of follow-up time in a ratio of 1:1, the matched case–control groups comprised 1150 patients with recurrent PUD and 1150 patients without recurrent PUD within 3 years of follow-up. More recurrent PUDs occurred in NSAID users than in the control group (75.30% versus 69.74%; *p* = 0.0028). Independent risk factors for recurrent PUD included patients using NSAIDs (adjusted OR (aOR): 1.34, *p* = 0.0040), *H. pylori* eradication (aOR: 2.73; *p* < 0.0001), concomitant H2 receptor antagonist (aOR: 1.85; *p* < 0.0001) and anti-coagulant (aOR: 4.21; *p* = 0.0242) use. Importantly, in the initial subgroup analysis, the risk ratio of recurrent PUD did not increase in NSAID users after *H. pylori* eradication compared with that in non-users (*p* = 0.8490) but a higher risk for recurrent PUD with the increased doses of NSAIDs without H. pylori eradication therapy (aOR: 1.24, *p* = 0.0424; aOR: 1.47, *p* = 0.0074; and aOR: 1.64, *p* = 0.0152 in the groups of ≤28, 29–83, and ≥84 cumulative defined daily doses, respectively). The current study suggested that *H. pylori* eradication therapy could decrease the risk of recurrent PUD among patients with high cumulative doses of NSAIDs.

## 1. Introduction

Nonsteroidal anti-inflammatory drug (NSAID)-induced ulcers are common in old patients, patients on multiple medications, patients with comorbidities, *Helicobacter pylori (H. pylori)* infection, and history of peptic ulcer [1,2]. A study has reported that 6.4%–11.8% of patients who were prescribed NSAIDs developed peptic ulcer disease (PUD), particularly NSAID-naïve patients (odds ratio (OR) = 0.26, 95% Confidence interval (CI): 0.14–0.49) compared with long term users of NSAIDs (OR = 0.74, 95% CI: 0.46–1.20) [3]. Vergara et al. added that *H. pylori* eradication therapy alone had no significant clinical benefit in chronic users in their study (OR = 0.95, 95%CI: 0.53–1.72) [4]. 

Both Asia-Pacific and American College of Gastroenterology Clinical guidelines state that patients with H. pylori infection who have used NSAIDs are at an increased risk for developing PUD [5,6]. Hence, patients with PUD who have used NSAIDs for primary prevention should be tested and treated for *H. pylori* infection. Regarding *H. pylori* eradication as a secondary preventive measure for PUD in NSAID users, Chan et al. observed that treatment with proton pump inhibitors (PPIs) was more beneficial in reducing recurrent bleeding over 6 months than *H. pylori* eradication alone. In clinical practice, it is important to prescribe concomitant PPIs for all NSAID users who have a medical record of PUD with complications [7].

It is well understood that by eliminating *H. pylori* for both primary and secondary prophylaxis effectively decrease ulcer risk among naïve NSAID users. However, it is not evidenced whether the success in killing of these bacteria can help in reducing the recurrence of peptic ulcers in long-term users of NSAIDs, particularly those with a medical record of PUD [8,9]. Importantly, very few studies have examined whether the eradication therapy affects the recurrence of PUD in patients receiving a high cumulative defined daily dose (cDDDs) of NSAIDs. Therefore, we used the Taiwan National Health Insurance Research database (NHIRD) in an attempt to clarify the impact of cumulative dosage of NSAIDs on recurrent peptic ulcers among chronic users after H. pylori eradication.

## 2. Methods

### 2.1. Study Population 

This study was reviewed and permitted by the Ethics Committee of Chang Gung Memorial Hospital in Taiwan (Institutional review board 201800321B0). Data required for the study were collected from the claims data recorded in Taiwan’s NHIRD, an anonymized dataset of one million randomly selected individuals from 1997 to 2013. Patients with original admission record of a major diagnosis of PUD or two outpatient visit PUD diagnoses >28 days apart (International Classification of Diseases, 9th Revision codes: 531x, 532x, 533x, and 534x) were selected. Patients with PUD (*n* = 203,407) were enrolled as shown in Figure 1. Eventually, 27,920 patients were enrolled for analysis after excluding 175,487 patients aged <18 years or patients who met PUD diagnostic criteria within 365 days before the index date; those who received NSAIDs, aspirin, PPIs, or H2 blockers within 180 days before the index date; those who received *H. pylori* eradication therapy before the index date; those who had bleeding varices; those with any malignancy who developed the disease before and after the index date; and those who developed serious infections [10,11] and suffered major traumas [12,13] after PUD diagnosis. 

A case-control study design of NSAID exposure and recurrent PUD was used in the PUD cohort. Recurrent PUD was defined as endoscopically proven PUD occurring 180 days’ post-index date (the first date of PUD diagnosis). The incident recurrent PUD risk was evaluated during 3 years of follow-up. Patients who developed recurrent PUD during the follow-up period were classified into the case group; patients without recurrent PUD (controls) were randomly selected by matching age/gender frequencies with patients in the case group in a 1:1 ratio. To avoid immortal time bias, the follow-up duration for patients in the case group was applied to the randomly selected matched controls. Immortal time refers to a span of time in the observation of the control group of patients during which they could not have developed recurrent PUD or could not have died. Following the follow-up time-matched controls, the misclassification of NSAIDs and relevant risk factor exposure between case and control groups could be avoided. Finally, two groups of patients were analyzed: recurrent PUD (*n* = 1185) and non-recurrent PUD (controls, *n* = 4395).

### 2.2. NSAIDs Exposure

A specific term known as the defined daily dose (DDD) has been suggested by the World Health Organization as a unit for quantifying a prescribed dose of medication anticipating the average prescription dose per day in adult population [14]. This allowed us to compare any class of NSAIDs on similar reference line: (sum of drug used)/(amount of drug in a DDD) = number of DDDs. cDDD refers to the total bare dose, assessed as the total of the allocated DDD of any NSAIDs, to equate the risk of PUD among a cohort. To further recognize the possible influence of the dose effect, we classified the NSAID dose into four sets in each group (0, ≤28, 29–83, and ≥84 cDDDs). Patients were considered as not taking any NSAIDs if the cDDD was zero.

### 2.3. Other Ulcerogenic Agents and Potential Confounders

We obtained the complete data from the database, which included the first and last prescription dates of all the ulcerogenic medications in between the follow-up dates during the follow-up period after the index hospitalization. These medications were aspirin, clopidogrel, dipyridamole, warfarin, ticlopidine, cilostazol, and Ginkgo biloba (cerenin^®^). Recorded comorbidities were simultaneously analyzed among those with a history of coronary artery disease, cerebrovascular accidents, hypertensive disease, diabetes mellitus, chronic obstructive lung disease, advanced chronic liver disease which included liver cirrhosis, and hyperlipidemia. Charlson comorbidity index was calculated as a potential confounding risk [15]. 

### 2.4. Helicobacter Pylori Therapy and Eradication

In the present study, we defined H. pylori-associated PUD as the use of any recorded *H. pylori* eradication therapy medications, such as a combination medication prescription of 7–14 days of any of the following antibiotics in the same medication order such as clarithromycin or metronidazole and amoxicillin or tetracycline, in addition to a PPIs or H2 blockers (H2RAs) [16,17,18]. Summary of *H. pylori* treatment regimens used in this study were summarized in Appendix A.

### 2.5. Statistical Analysis 

All statistical analyses were completed by using software package SAS version 9.3 (SAS Institute Inc., Cary, NC, USA, 2013) with descriptive measurements for all variables. Continuous statistics were expressed as mean (standard deviation) and median (interquartile range). Categorical statistics were expressed as actual frequencies and percentages. Unpaired Student’s t-test and chi-square analysis of contingency tables for continuous and insignificant variables were applied to compare baseline characteristics. Logistic regression modeling was used to fix the potentially relevant issues influencing the outcome with modifications in the multivariate analysis. A *p*-value < 0.05 was considered significant. 

## 3. Results

Among the study patients, there were 1185 patients with recurrent PUD (case group) and 4395 patients without recurrent PUD (control group). After matching for age/gender frequencies and the length of follow-up time in a 1:1 ratio, the matched case-control groups comprised 1150 patients with recurrent PUD and 1150 patients without recurrent PUD within 3 years of follow-up (mean follow-up time: 20.65 (8.77) months).

Table 1 presents a comparison of patient characteristics between the case and control groups. The proportion of NSAID users was higher in the case (recurrent PUD) group than in the control group in both primary (74.85% versus 63.32%; *p* < 0.0001) and matched (75.30% versus 69.74%; *p* = 0.0028) cohorts. In the matched case–control group, *H. pylori* eradication therapy (8.61% versus 3.48%; *p* < 0.0001), concomitant PPI users (22.43% versus 17.22%; *p* = 0.0017); and H2RA users (26.87% versus 17.57%; *p* < 0.0001) were significantly higher in the case (recurrent PUD) group than in the control group.

Multivariate analysis (Table 2) showed that the independent risk factors for recurrent PUD were patients using NSAIDs (aOR: 1.34; *p* = 0.0040), those with *H. pylori* eradication (aOR: 2.73; *p* < 0.0001), and those with concomitant H2RA (aOR: 1.85; *p* < 0.0001) and anti-coagulant (aOR: 4.21, *p* = 0.0242) use, but not with PPI use (aOR: 1.13; *p* = 0.3022). The risk for recurrent PUD increased with the increasing NSAID dosage (aOR: 1.24, *p* = 0.0430; aOR: 1.52, *p* = 0.0043; and aOR: 1.67, *p* = 0.0179 in the groups of ≤28, 29–83, and ≥84 cDDDs, respectively). The use of diclofenac, piroxicam, and sulindac was a significant risk factor for recurrent PUD (aOR: 1.33, *p* = 0.0042; aOR: 1.59, *p* = 0.0346; and aOR: 1.59, *p* = 0.0367, respectively).

In the subgroup analysis, the risk ratio of recurrent PUD did not increase in NSAID users after *H. pylori* eradication compared with that in non-users (*p* = 0.8490). However, there was a progressively higher risk for recurrent PUD with the increasing doses of NSAIDs without *H. pylori* eradication therapy (aOR: 1.24, *p* = 0.0424; aOR: 1.47, *p* = 0.0074; and aOR: 1.64, *p* = 0.0152 in the groups of ≤28, 29–83, and ≥84 cDDDs, respectively) (Table 3).

## 4. Discussion

According to published guidelines, *H. pylori* eradication therapy alone is suboptimal for the secondary prophylaxis of ulcers or PUD bleeding in patients who have continuously been prescribed NSAIDs [5,6]. Instead, concomitant use of PPIs or switching to cyclooxygenase-2 (COX-2) inhibitors or a combination of COX-2 inhibitors and PPIs is still recommended [19,20]. Furthermore, it was found that the optimal gastrointestinal protection could be achieved by using selective COX-2 inhibitors with PPIs [21]. In this study, the risk of recurrent PUD was high in patients using concomitant H2RAs (aOR: 1.85; *p* < 0.0001) but not in those using PPIs (aOR: 1.13; *p* = 0.3022). Therefore, the protective efficacy of concomitant PPIs for recurrent PUD is better than that of H2RAs. This result is similar to another database systematic review study [22], but a double-dose H2RA may have considerably more beneficial outcome than PPIs in reducing the risk of duodenal and gastric ulcers (relative risk (RR) = 0.44 and 0.40, respectively). 

At the first glance of the initial analysis, *H. pylori* eradication was found to be a factor for recurrent PUD (aOR: 2.73; *p* < 0.0001). However, in the subgroup analysis, the risk ratio of recurrent PUD did not increase after *H. pylori* eradication in NSAID users compared with that in non-users (*p* = 0.8490). In patients without H. pylori eradication therapy as shown in Table 3, there was a progressively higher risk for recurrent PUD with the increasing doses of NSAIDs (aOR: 1.24, *p* = 0.0424; aOR: 1.47, *p* = 0.0074; and aOR: 1.64, *p* = 0.0152 in the groups of ≤28, 29–83, ≥84 cDDDs, respectively). This implied that *H. pylori* eradication therapy decreased the risk of recurrent ulcers during the prolonged NSAID use. Although the eradication alone was suboptimal in reducing recurrent PUD, it was still needed in NSAID users who required prolonged medications. According to some studies in Taiwan [23,24], approximately 53.9% (95% CI, 36.6–71.2) patients with H. pylori infection were not evaluated for *H. pylori* status. In other words, in H. pylori-positive patients without the eradication therapy, the risk of recurrent PUD may increase with the increase in the dosage of NSAIDs used. Our study was similar to the previous population-based studies from Spain [25,26,27], taking into consideration the daily dosage, the RR was 2.79 (95% CI, 2.17–3.58) for the NSAIDs users at low or medium doses compared to 5.36 (95% CI, 4.57–6.29) among those receiving high doses of medications [28]. In real-world clinical practice, poor adherence to guideline recommendations on the issue of the risk of PUD bleeding without prophylactic PPI prescription is common among NSAID users. Even in this national population study, the rate of concomitant PPI use was only 22.45% before matching. In a Turkish population, Dincer et al. reported that only 25% of NSAID users with a previous history of peptic ulcer bleeding received a PPI prophylaxis [29].

The strength of this study was that it was a large sample size population-based case controlled study and quantify the level of drug exposure to the association of recurrent PUD among chronic users after H. pylori eradication by the concept of cDDD. Similarly, it still came across some limitations. First, the number of cases with *H. pylori* infection was low, owing to strict exclusion criteria, and, consequently, not enough to analyze the NSAID dose effect in *H. pylori* eradication group. Another major limitation of our study data retrieved from NHIRD was the lack of anthropometric data and individual status, such as atrophic grades of gastric mucosa, drug compliance, *H. pylori* eradication rate, and antibiotic resistance. This limitation is due to the inherent shortcomings of the administrative database. Drug compliance is an important factor in the assessment of NSAIDs use with gastric bleeding, but compliance to any medications is not available in the database of health insurance system. This was overcome by using the concept of cDDD, which designated the total exposed dosage assessed as the total of allotted DDD of any NSAIDs to compare the risk of PUD among them. Eventually, we were able to analyze cDDD by using the Taiwan NHI program. This was because it was a compulsory, third party payer insurance program, and provides comprehensive health services such as procedures, medications in inpatient, outpatient and emergency department for almost 100% of the Taiwanese. Third, Bytzer et al. reported that low socioeconomic class was a risk factor for peptic ulcer disease [30]. However, urbanization or income levels were not assessed in the present study. Finally, this was a retrospective study rather than a prospective intervention trial, limiting the ability to show cause and effect.

## 5. Conclusions

The risk of recurrent PUD increases progressively in patients with high cumulative doses of NSAIDs. *H. pylori* eradication therapy could reduce the risk of recurrent PUD among NSAID users who needed prolonged treatment durations. *H. pylori* eradication alone is suboptimal for NSAID users with a medical record of PUD and concomitant PPI treatment is required in such patients.

## Figures and Tables

**Figure 1 jcm-08-01722-f001:**
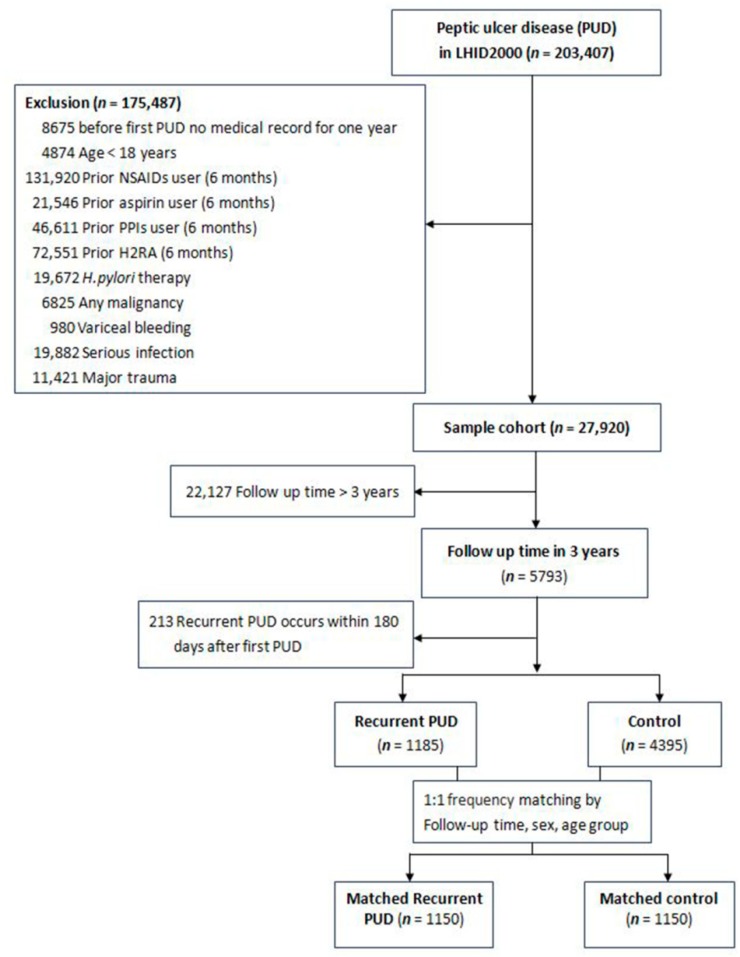
Flowchart depicting participant selection.

**Table 1 jcm-08-01722-t001:** Characteristics between patients with and without recurrent peptic ulcer disease.

Variables	Primary Case-Control Group				Matched Case-Control Group			
	All (*n* = 5580)	Recurrent PUD (*n* = 1185)	Control (*n* = 4395)	*p* Value	All (*n* = 2300)	Recurrent PUD (*n* = 1150)	Control (n = 1150))	*p* Value
**Follow up period (months)**	18.33 ± 10.66	20.47 ± 8.79	17.75 ± 11.04	<0.0001	20.65 ± 8.77	20.65 ± 8.77	20.65 ± 8.77	1.0000
**NSAID use**				<0.0001				0.0028
DDD = 0	1910	298(25.15)	1612(36.68)		632	284(24.70)	348(30.26)	
DDD > 0	3670	887(74.85)	2783(63.32)		1668	866(75.30)	802(69.74)	
**NSAID cDDD group**				<0.0001				0.0156
DDD = 0	1910	298(25.15)	1612(36.68)		632	284(24.70)	348(30.26)	
0 < DDD ≤ 28	2736	632(53.33)	2104(47.87)		1203	614(53.39)	589(51.22)	
28 < DDD ≤ 84	667	183(15.44)	484(11.01)		335	180(15.65)	155(13.48)	
84 < DDD	267	72(6.08)	195(4.44)		130	72(6.26)	58(5.04)	
**HP eradication therapy**	263	102(8.61)	161(3.66)	<0.0001	139	99(8.61)	40(3.48)	<0.0001
**Gender**				<0.0001				1.0000
Female	2623	459(38.73)	2164(49.24)		912	456(39.65)	456(39.65)	
Male	2957	726(61.27)	2231(50.76)		1388	694(60.35)	694(60.35)	
**Age group (years)**				<0.0001				1.0000
18–30	655	118(9.96)	537(12.22)		224	112(9.74)	112(9.74)	
31–40	900	193(16.29)	707(16.09)		384	192(16.70)	192(16.70)	
41–50	1123	288(24.30)	835(19.00)		566	283(24.61)	283(24.61)	
51–60	1086	254(21.43)	832(18.93)		480	240(20.87)	240(20.87)	
61–70	724	164(13.84)	560(12.74)		310	155(13.48)	155(13.48)	
>70	1092	168(14.18)	924(21.02)		336	168(14.61)	168(14.61)	
**Comorbidity**								
Acute myocardial infarction	30	3(0.25)	27(0.61)	0.1313	8	3(0.26)	5(0.43)	0.4787
Congestive heart failure	111	5(0.42)	106(2.41)	<0.0001	25	5(0.43)	20(1.74)	0.0026
Peripheral vascular disease	26	0(0.00)	26(0.59)	0.0080	7	0(0.00)	7(0.61)	0.0081
Cerebral vascular accident	283	41(3.46)	242(5.51)	0.0044	86	40(3.48)	46(4.00)	0.5096
Dementia	69	8(0.68)	61(1.39)	0.0488	16	8(0.70)	8(0.70)	1.0000
Pulmonary disease	470	93(7.85)	377(8.58)	0.4221	184	93(8.09)	91(7.91)	0.8778
Connective tissue disorder	29	7(0.59)	22(0.50)	0.7017	9	7(0.61)	2(0.17)	0.0949
Liver disease	514	137(11.56)	377(8.58)	0.0016	216	127(11.04)	89(7.74)	0.0066
Diabetes	485	78(6.58)	407(9.26)	0.0037	169	77(6.70)	92(8.00)	0.2306
Diabetes complications	119	18(1.52)	101(2.30)	0.0995	35	17(1.48)	18(1.57)	0.8647
Paraplegia	18	2(0.17)	16(0.36)	0.2928	5	2(0.17)	3(0.26)	0.6544
Renal disease	188	35(2.95)	153(3.48)	0.3716	66	34(2.96)	32(2.78)	0.8027
Severe liver disease	25	2(0.17)	23(0.52)	0.1048	4	2(0.17)	2(0.17)	1.0000
HIV	4	0(0.00)	4(0.09)	0.2989	2	0(0.00)	2(0.17)	0.1571
Hypertension	1073	205(17.30)	868(19.75)	0.0575	393	197(17.13)	196(17.04)	0.9558
Hyperlipidemia	455	80(6.75)	375(8.53)	0.0467	177	79(6.87)	98(8.52)	0.1372
**Medication use in the baseline period**								
Antiplatelet	236	44(3.71)	192(4.37)	0.3197	85	44(3.83)	41(3.57)	0.7402
Anti-coagulants	19	2(0.17)	17(0.39)	0.2529	4	2(0.17)	2(0.17)	1.0000
ACEI/ARB	519	105(8.86)	414(9.42)	0.5565	208	100(8.70)	108(9.39)	0.5608
SSRIs	18	2(0.17)	16(0.36)	0.2928	5	2(0.17)	3(0.26)	0.6544
**Concomitant medications during follow-up**								
PPIs	1036	266(22.45)	770(17.52)	0.0001	456	258(22.43)	198(17.22)	0.0017
H2RA	1017	317(26.75)	700(15.93)	<0.0001	511	309(26.87)	202(17.57)	<0.0001
Antiplatelet	560	129(10.89)	431(9.81)	0.2724	258	126(10.96)	132(11.48)	0.6918
Anti-coagulants	44	14(1.18)	30(0.68)	0.0849	20	14(1.22)	6(0.52)	0.0724
Cerenin^®^	40	9(0.76)	31(0.71)	0.8445	18	9(0.78)	9(0.78)	1.0000
Aspirin	269	67(5.65)	202(4.60)	0.1314	133	64(5.57)	69(6.00)	0.6551
SSRIs	116	22(1.86)	94(2.14)	0.5456	50	21(1.83)	29(2.52)	0.2527

Abbreviations: PUD, peptic ulcer disease; NSAIDs, non-steroidal anti-inflammatory drugs; cDDD, cumulative defined daily doses; HP, *Helicobacter pylori*; HIV, human immunodeficiency virus; ACEI/ARB, angiotension-converting enzyme *inhibitor/* angiotension receptor blocker, PPIs, proton pump inhibitors; H2RA, H2 receptor antagonists; Cerenin^®^, Ginkgo biloba leaves extract; SSRIs, selective serotonin reuptake inhibitors; SD (standard deviation).

**Table 2 jcm-08-01722-t002:** Factors associated with recurrent PUD occurrence in matched case-control group.

Variables	Matched Case-Control Group		
	aOR	95% CI	*p*-Value
**NSAID cDDD group (ref: cDDD = 0)**			
cDDD ≤ 28	1.24	(1.01–1.52)	0.0430
28 < cDDD ≤ 84	1.52	(1.14–2.02)	0.0043
84 < cDDD	1.67	(1.09–2.56)	0.0179
HP eradication therapy	2.73	(1.80–4.15)	<0.0001
**Kinds of NSAIDs**			
diclofenac	1.33	(1.09–1.61)	0.0042
mefenamic acid	1.05	(0.86–1.28)	0.6379
ibuprofen	0.98	(0.78–1.22)	0.8409
ketorolac	0.95	(0.73–1.23)	0.7039
ketoprofen	1.27	(0.90–1.79)	0.1696
naproxen	1.38	(0.98–1.93)	0.0630
acemetacin	0.86	(0.61–1.21)	0.3839
meloxicam	1.15	(0.76–1.76)	0.5045
celecoxib	0.80	(0.50–1.28)	0.3477
piroxicam	1.59	(1.03–2.44)	0.0346
indometacin	1.04	(0.65–1.68)	0.8585
flubiprofen	1.15	(0.75–1.76)	0.5224
sulindac	1.59	(1.03–2.45)	0.0367
**Comorbidity**			
Acute myocardial infarction	0.54	(0.11–2.71)	0.4517
Congestive heart failure	0.20	(0.07–0.61)	0.0048
Cerebral vascular accident	0.85	(0.52–1.40)	0.5339
Dementia	1.41	(0.47–4.22)	0.5425
Pulmonary disease	1.09	(0.79–1.51)	0.6021
Connective tissue disorder	3.56	(0.69–18.29)	0.1276
Liver disease	1.47	(1.09–2.00)	0.0128
Diabetes	0.75	(0.52–1.10)	0.1407
Diabetes complications	0.89	(0.41–1.95)	0.7742
Paraplegia	1.59	(0.24–10.53)	0.6336
Renal disease	0.97	(0.55–1.71)	0.9150
Severe liver disease	0.45	(0.06–3.45)	0.4414
Hypertension	1.16	(0.84–1.61)	0.3554
Hyperlipidemia	0.78	(0.55–1.11)	0.1668
**Medications use in the baseline period**			
Antiplatelet	1.35	(0.78–2.35)	0.2802
Anti-coagulants	2.22	(0.14–35.03)	0.5703
ACEI/ARB	0.89	(0.60–1.34)	0.5849
SSRIs	0.87	(0.13–5.73)	0.8878
**Concomitant medications during follow up**			
PPIs	1.13	(0.89–1.44)	0.3022
H2RA	1.85	(1.47–2.32)	<0.0001
Antiplatelet	0.65	(0.41–1.03)	0.0688
Anti-coagulants	4.21	(1.21–14.68)	0.0242
Cerenin^®^	1.08	(0.40–2.91)	0.8794
Aspirin	1.30	(0.76–2.23)	0.3382
SSRIs	0.63	(0.35–1.15)	0.1320

In the models comparing cDDD > 0 to cDDD = 0, primary cohort: aOR = 1.69 (1.45–1.96), *p* < 0.0001; matched cohort: aOR = 1.34 (1.10–1.64), *p* = 0.004. Abbreviations: PUD, peptic ulcer disease; NSAIDs, non–steroidal anti-inflammatory drugs; cDDD, cumulative defined daily doses; HP, *Helicobacter pylori*; ACEI/ARB, angiotension-converting enzyme inhibitor/angiotension receptor blocker, PPIs, proton pump inhibitors; H2RA, H2 receptor antagonists; Cerenin^®^, Ginkgo biloba leaves extract; SSRIs, selective serotonin reuptake inhibitors; aOR, adjusted odds ratio; CI, confidence interval.

**Table 3 jcm-08-01722-t003:** Factors associated with recurrent PUD occurrence between with and without *H pylori* therapy groups in the matched case-control groups.

	*H pylori* Therapy			Without *H pylori* Therapy		
	aOR	95% CI	*p*-Value	aOR	95% CI	*p*-Value
NSAID use						
cDDD > 0 (ref:DDD = 0)	1.09	(0.43–2.77)	0.8490	1.31	(1.08–1.59)	0.0068
**NSAID cDDD group (ref: cDDD = 0)**						
cDDD ≤ 28		-		1.24	(1.01–1.52)	0.0424
28 < cDDD ≤ 84		-		1.47	(1.11–1.94)	0.0074
84 < cDDD		-		1.64	(1.10–2.45)	0.0152
**Comorbidity**						
Acute myocardial infarction		-		0.48	(0.10–2.32)	0.3591
Congestive heart failure		-		0.23	(0.08–0.71)	0.0104
Cerebral vascular accident	0.52	(0.06–4.75)	0.5597	0.83	(0.50–1.39)	0.4737
Dementia		-		1.26	(0.42–3.74)	0.6772
Pulmonary disease	2.57	(0.28–23.39)	0.4030	0.99	(0.72–1.36)	0.9463
Connective tissue disorder		-		3.31	(0.67–16.41)	0.1420
Liver disease	3.84	(0.43–34.17)	0.2283	1.50	(1.11–2.02)	0.0083
Diabetes	0.72	(0.13–4.06)	0.7049	0.80	(0.55–1.17)	0.2567
Diabetes complications		-		0.85	(0.39–1.88)	0.6929
Paraplegia		-		1.12	(0.17–7.36)	0.9042
Renal disease		-		0.86	(0.49–1.51)	0.6063
Severe liver disease		-		0.55	(0.07–4.24)	0.5704
Hypertension	0.67	(0.15–3.01)	0.6036	1.17	(0.86–1.59)	0.3299
Hyperlipidemia	1.51	(0.26–8.88)	0.6470	0.74	(0.53–1.05)	0.0886
**Medications use in the baseline period**						
Antiplatelet	0.79	(0.11–5.62)	0.8136	1.40	(0.80–2.45)	0.2390
Anti-coagulants		-		0.64	(0.05–9.04)	0.7416
ACEI/ARB	1.57	(0.23–10.80)	0.6474	0.87	(0.59–1.30)	0.5029
SSRIs		-		0.87	(0.14–5.45)	0.8850
**Concomitant medications during follow up**						
PPIs	0.38	(0.17–0.86)	0.0199	1.24	(0.98–1.58)	0.0735
H2RA	1.89	(0.70–5.09)	0.2066	1.74	(1.39–2.18)	<0.0001
Anti-platelet agents	2.98	(0.55–16.11)	0.2041	0.65	(0.41–1.02)	0.0598
Anti-coagulants		-		6.34	(1.59–25.32)	0.0089
Cerenin^®^		-		0.91	(0.33–2.50)	0.8529
Aspirin		-		1.17	(0.68–2.03)	0.5663
SSRIs		-		0.59	(0.32–1.08)	0.0882

“-”: Numbers of event was too small in the variable group to calibrate odds ratio. Abbreviations: PUD, peptic ulcer disease; NSAIDs, non-steroidal anti-inflammatory drugs; cDDD, cumulative defined daily doses; HP, *Helicobacter pylori*; ACEI/ARB, angiotension-converting enzyme inhibitor/angiotension receptor blocker, PPIs, proton pump inhibitors; H2RA, H2 receptor antagonists; Cerenin^®^, Ginkgo biloba leaves extract; SSRIs, selective serotonin reuptake inhibitors; aOR, adjusted odds ratio; CI, confidence interval.

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
