# Peer review of "Risk of Recurrent Peptic Ulcer Disease in Patients Receiving Cumulative Defined Daily Dose of Nonsteroidal Anti-Inflammatory Drugs"

_jcm, 2019, doi:10.3390/jcm8101722_

Round 1

Reviewer 1 Report

Comments to the author:

This is a study aimed to clarify the risks of recurrence PUD in those who had been treated with NSAIDs based on a nationwide population dataset in Taiwan. It is known that they have a big database of health insurance system. The data of this paper was coming from the database. The paper was well written and easy to follow. It is of importance to clarify factors of PUD recurrence specifically patients with NSAIDs. I do, however, have several concerns about the study that need to be addressed.

Major comments:

It has already been evidenced that there are significant differences of PUD risks among NSAIDs. The authors should analyze the risks of each NSAIDs risks and dose of each NSAIDs, because it is very important for clinicians to select NSAIDs in the clinical setting. The authors should show clinical comparisons of atrophic grades in enrolled patients because atrophic grades and patterns are evidenced to be associated with PUD. The authors should think about compliance of the patient for medical treatment because the results were from insurance system database. The authors should examine and stratify the results according to urbanization level or income levels.

Minor comments:

Abbreviations must be spelled out when they are firstly appeared.

Author Response

Dear Reviewer,

Thank you for your constructive comments.

We have revised the paper and response to your comments point-by-point as attached file.

Should there be any further comments, please let us know

Best regards,

Prof. Chuah

This is a study aimed to clarify the risks of recurrence PUD in those who had been treated with NSAIDs based on a nationwide population dataset in Taiwan. It is known that they have a big database of health insurance system. The data of this paper was coming from the database. The paper was well written and easy to follow. It is of importance to clarify factors of PUD recurrence specifically patients with NSAIDs. I do, however, have several concerns about the study that need to be addressed.

Major comments:

It has already been evidenced that there are significant differences of PUD risks among NSAIDs. The authors should analyze the risks of each NSAIDs risks and dose of each NSAIDs, because it is very important for clinicians to select NSAIDs in the clinical setting.

Reply: Thank you very much for this insightful comment.

We have reanalyzed adjusted odds ratios for recurrent peptic ulcer with each NSAID and have presented the results in Table 2 (Matched case–control group). The use of diclofenac, piroxicam, and sulindac was a significant risk factor for recurrent PUD (aOR: 1.33, p = 0.0042; aOR: 1.59, p = 0.0346; and aOR: 1.59, p = 0.0367, respectively). We have added the following sentence to the Results section:

Revised sentence (Page 9, line 19, page 10, lines 1-2 )

The use of diclofenac, piroxicam, and sulindac was a significant risk factor for recurrent PUD (aOR: 1.33, p = 0.0042; aOR: 1.59, p = 0.0346; and aOR: 1.59, p = 0.0367, respectively).

The authors should show clinical comparisons of atrophic grades in enrolled patients because atrophic grades and patterns are evidenced to be associated with PUD. The authors should think about compliance of the patient for medical treatment because the results were from insurance system database. The authors should examine and stratify the results according to urbanization level or income levels.

Reply: Thank you very much for this important comment.

We agree that drug compliance is an important factor in association of NSAIDs use with gastric bleeding. As the title of this paper indicated, “The risks of recurrence peptic ulcer disease in patients who receive cumulative defined daily dose of non-steroidal anti-Inflammatory drugs” However, compliance to any medications is not available in the database of health insurance system. Alternately, a specific term known as the defined daily dose (DDD) has been suggested by the World Health Organization as a unit in quantifying a prescribed dose of medication anticipating average preservation dose per day in adult patients [14]. This allowed us to compare any classes of NSAIDs on the same reference line: (total amount of drug) / (amount of drug in a DDD) = number of DDDs. On the other hand, cumulative defined daily dose (cDDD) designates the total exposed dosage which was assessed as the total of allotted DDD of any NSAIDs to compare the risk of PUD among them.

Therefore, we were able to analyze cDDD which was employed to quantify the level of drug exposure to the association PUD by using the Taiwan NHI program. This was because it is a compulsory, third party payer insurance program, and provides comprehensive health services such as procedures, medications in inpatient, outpatient and emergency department for almost 100% of the Taiwanese.

Revised sentences (Page 12, lines 11-19, and Page 13, lines 1-3)

Another major limitation of our study data retrieved from NHIRD was the lack of anthropometric data and individual status, such as atrophic grades of gastric mucosa, drug compliance, H. pylori eradication rate, and antibiotic resistance. This limitation is due to the inherent shortcomings of the administrative database. Drug compliance is an important factor in the assessment of NSAID use with gastric bleeding, but compliance to any medications is not available in the database of health insurance system. This was overcome by using the concept of cDDD which designated the total exposed dosage assessed as the total of allotted DDD of any NSAIDs to compare the risk of PUD among them. Eventually, we were able to analyze cDDD by using the Taiwan NHI program. This was because it was a compulsory, third party payer insurance program, and provides comprehensive health services such as procedures, medications in inpatient, outpatient and emergency department for almost 100% of the Taiwanese.

The authors should examine and stratify the results according to urbanization level or income levels.

Bytzer et al reported that low socioeconomic class was a risk factor for peptic ulcer disease [30]. However, urbanization or income levels were not examined in this study. It was the limitation of current study

Revised sentence: (Page 13, lines 3-7)

Third, Bytzer et al reported that low socioeconomic class was a risk factor for peptic ulcer disease [30]. However, urbanization or income levels were not assessed in the present study. Finally, this was a retrospective study rather than a prospective intervention trial, limiting the ability to show cause and effect.

Minor comments:

Abbreviations must be spelled out when they are firstly appeared. 

Reply: Thank you very much for reminding this. We have made changes accordingly.

Reviewer 2 Report

The study is interesting and the sample size is adequate. The main limitation is its retrospective design. However, its publication can be of interests, particularly in relation to  the role of H. pylori eradication in the prevention of peptic ulcer in NSAID users.

This reviewer suggests the Authors to simplify the tables for the reader

English language must be reviewed

Author Response

Dear reviewer,

Thanks for your comments.

We have revised the manuscript and provide a point-by-point response to the comments

Should there be further comments, please let us know!

Best regards,

Prof. Chuah

Reviewer 3 Report

The context of abstract is not adequate for publication. Helicobacter pylori therapy and eradication. This section should be better described or summarized in additional table. How did presence of infection was checked? (methods). How did effectiveness of therapy was monitored? Division of patients according to therapy should be added. How did authors check resistance to  applied antibiotics? It is very common problem that can influenced on relapsed of diseases. H.pylori should be written by italic.

Author Response

Dear Reviewer,

Thank you for your constructive comments.

We have revised the manuscript and provide a point-by-point response to your comments

Should there be further comments, please keep us informed\

Best regards,

Prof. Chuah

Round 2

Reviewer 1 Report

The authors have replied my questions.